# A Simple Technique to Enable Saliency Methods to Pass the Sanity Checks

## Abstract

*Saliency methods* attempt to explain a deep net's decision by assigning a *score* to each feature/pixel in the input, often doing this credit-assignment via the gradient of the output with respect to input. Recently Adebayo et al. (2018) questioned the validity of many of these methods since they do not pass simple *sanity checks*, which test whether the scores shift/vanish when layers of the trained net are randomized, or when the net is retrained using random labels for inputs.

We propose a simple fix to existing saliency methods that helps them pass sanity checks, which we call *competition for pixels*. This involves computing saliency maps for all possible labels in the classification task, and using a simple competition among them to identify and remove less relevant pixels from the map. Section 3 provides a theoretical justification to support our method and indicates when it can be applied, and its performance is empirically demonstrated on several popular methods.

## 1 Introduction

*Saliency methods* attempt to explain a deep net's decision to humans by assigning a *score* to each feature/pixel in the input, often doing this credit-assignment via the gradient of the output with respect to input (from now on refered to as just "gradient"). Here we are interested in tasks involving multiclass classification, and for simplicity the exposition will assume the input is an image. Then a saliency method assigns scores to input pixels, which are presented as a heat map. (Extensions of these ideas to higher-level features of the net will not be discussed here.)

While gradient alone is often too noisy, it as well as related notions are the basis of other more successful methods. In **Gradient ⊙ Input** (Shrikumar et al., 2017) the pixel score is the product of the corresponding coordinate of gradient vector with the pixel value. **Layer-wise Relevance Propagation (LRP)** (Bach et al., 2015) uses a back-propagation technique where every node in the deep net receives a share of the output which it distributes to nodes below it. This happens all the way to the input layer, whereby every pixel gets assigned a share of the output, which is its score. Another rule **Deep-Lift** (Shrikumar et al., 2016) does this in a different way and is related to Shapley values of cooperative game theory. **DASP** Ancona et al. (2019) is a state of the art method that performs an efficient approximation of the Shapley values. The perceived limitations of these methods in turn motivated a long list of new ones. Montavon et al. (2018) provides a survey of existing methods, and brief discussion is presented in Section 2.

The focus of the current paper is an evaluation of saliency methods called *sanity checks* in Adebayo et al. (2018). This involves randomizing the model parameters or the data labels (see Section 2 for details). The authors show that maps produced using corrupted parameters and data are often difficult to visually distinguish from those produced using the original parameters and data. The authors concluded that "*...widely deployed saliency methods are independent of both the data the model was trained on, and the model parameters.*"

The current paper shows how to pass sanity checks via a simple modification to existing methods: *Competition for pixels*. Section 3 motivates this idea by pointing out a significant issue with previous methods: they produce saliency maps for a chosen output (label) node using gradient information only for that node while ignoring the gradient information from the other (non-chosen) outputs. To incorporate information from non-chosen labels/outputs in the multiclass setting we rely on an

axiom called *completeness* satisfied by many saliency methods, according to which the sum of pixel scores in a map is equal to the value of the chosen node (see Section 3). Existing methods design saliency maps for all outputs and the map for each label satisfies completeness. One can then view the various scores assigned to a single pixel as its "votes" for different labels. The competition idea is roughly to zero out any pixel whose vote for the chosen label was lower than for another (non-chosen) label. Section 4 develops theory to explain why this modification helps pass sanity checks in the multi-class setting, and yet produces maps not too different from existing saliency maps. It also introduces a notion called *approximate completeness* and suggests that it is both a reasonable alternative to completeness in practice, and also allows our analysis of the competition idea to go through. We the present an new empirical finding that saliency methods that were not designed to satisfy completeness in practice seem to satisfy approximate completeness anyway. This may be relevant for future research in this area.

Section 5 reports experiments applying the competition idea to three well-regarded methods, **Gradient ⊙ Input**, **LRP**, and **DASP**, and shows that they produce sensible saliency maps while also passing the sanity checks. List of testbeds and methods is largely borrowed from Adebayo et al. (2018), except for inclusion of **DASP**, which draws inspiration from cooperative game theory.

## 2 RELATED WORK

Adebayo et al. (2018) and Montavon et al. (2018) provide surveys of saliency methods. Brief descriptions of some methods used in our experiments appear in Appendix Section 7.1. Here we briefly discuss the issue most relevant to the current paper, which is the interplay between tests/evaluations of saliency methods and principled design of new methods.

Controversies here often boil down to interpretations of the word "saliency," which method designers have sought to codify via axioms. (*Completeness* is a simple example.) Evaluations then suggest that the axioms fail to ensure other desirable properties. To give an example, does the map change significantly if we blank out or modify a portion of the image that humans find insignificant, as depicted in Kim et al. (2019) ? But in such cases it is sometimes unclear if the failure is due to the method alone, or traces to other unsolved issues in deep learning related to distribution shift, domain adaptation, adversarial examples, etc.

ROAR evaluation Hooker et al. (2018) greys out pixels/features found informative by a saliency method, and retrains the classifier on these modified inputs. The method is considered low quality if the accuracy drop is less than greying out the same fraction of randomly-chosen pixels. Many popular methods (including some tested here) do poorly on this, and a subset of ensemble methods outperform a random assignment of saliency scores. But clearly this evaluation is using a different definition of saliency than the axiomatic approaches. The axiomatic approaches seek to find a set of pixels that are *sufficient* to justify the output label. Since real-life images have high redundancy, multiple sets of pixels could be sufficient to justify the label. ROAR defines the goal as identifying every pixel that is potentially relevant, and then ensembling would clearly help. (To see this, imagine taking any dataset and duplicating each feature $k$ times.)

The current paper sidesteps some of these controversies by focusing solely on the *sanity checks* evaluation of Adebayo et al. (2018), and their exact testbed and framework. At first sight the problems uncovered by the sanity checks appear quite serious. While the distribution-shift based evaluations were asking "*Does the map shift too much upon changes to the input that humans find insignificant?*," the sanity checks asks the simpler question "*Does the map fail to shift when the model/labels are dramatically changed to become nonsensical?*" So it is surprising that most methods already failed the following sanity checks (the authors suggest there could be others).

**The model parameter randomization test.** According to the authors, this "compares the output of a saliency method on a trained model with the output of the saliency method on a randomly initialized untrained network of the same architecture." The saliency method fails if the maps are similar for trained models and randomized models. The randomization can be done in a cascading or layerwise fashion.

**The data randomization test** "compares a given saliency method applied to a model trained on a labeled data set with the method applied to the same model architecture but trained on a copy of the data set in which we randomly permuted all labels." Clearly the model in the second case has learnt

no useful relationship between the data and the labels and does not generalize. The saliency method fails if the maps are similar in the two cases on test data.

To the best of our knowledge, no subsequent paper has designed reasonable saliency methods that pass the sanity checks.

## 3 ADDING COMPETITION

The idea of competition suggests itself naturally when one examines saliency maps produced using *all* possible labels/logits in a multiclass problem, rather than just the chosen label. Figure 1 shows some **Gradient⊙Input** maps produced using AlexNet trained on MNIST, where the first layer was modified to accept one color channel instead of 3. Notice: *Many pixels found irrelevant by humans receive heat (i.e., positive value) in all the maps, and many relevant pixels receive heat in more than one map.* Our experiments showed similar phenomenon on more complicated datasets such as ImageNet. Figure 1 highlights an important point of Adebayo et al. (2018), which is that many saliency maps pick up a lot of information about the input itself —e.g., presence of sharp edges– that may be incidental to the final classification. Furthermore, these incidental features can survive during the various randomization checks, leading to failure in the sanity check. Thus it is a natural idea to create a saliency map by *combining* information from all labels, in the process filtering out or downgrading the importance of incidental features.

Suppose the input is $x$ and the net is solving a $k$-way classification. We assume a standard softmax output layer whose inputs are $k$ logits, one per label. Let $x$ be an input, $y$ be its label and $\ell_y$ denote the corresponding logit.

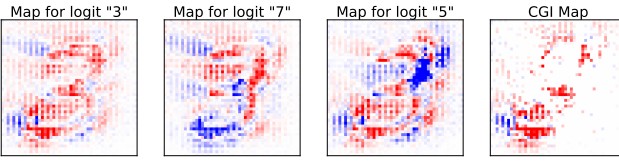

Figure 1: Heatmap of **Gradient ⊙ Input** saliency maps produced by various logits of a deep net trained on MNIST. Red denotes pixels with positive values and Blue denotes negative values. The input image is of the number $3$ , which is clearly visible in all maps. Note how maps computed using logits/labels "7" and " 5" assign red color (resp., blue color) to pixels that would have been expected to be present (resp., absent) in those digits. The last figure shows the map produced using our CGI method.

Usually prior methods do not combine information from maps from different labels, but we wish to do so to design a simple competition among labels for pixels. *A priori* it can be unclear how to compare scores across labels, since this could end up being an "apples vs oranges" comparison due to potentially different scaling. However, prior work Sundararajan et al. (2017) has identified a property called *completeness*: this requires that the sum of the pixel scores is exactly the logit value. For example, **Gradient ⊙ Input** satisfies completeness exactly for ReLU nets with zero bias[1]. (Recall that the ReLU function with *bias a* is $ReLU(z, a) \max\{z - a, 0\}$. ) Ensuring completeness for nonzero bias requires more computationally expensive operations.

**Enter competition.** Completeness (and also *approximate completeness*, a property explained below in Section 4.1) allows an apples-to-apples comparison of saliency scores from different labels, and to view them as "votes" for a label. Now consider the case where $y$ is the label predicted by the net for input $x$. Suppose pixel $i$ has a positive score for label $y$ and an even more positive score for label $y_1$. This pixel contributes positively to both logit values. But remember that since label $y_1$ was not predicted by the net as the label, the logit $\ell_{y_1}$ is *less* than than logit $\ell_y$, so the contribution of pixel $x_i$'s "vote" to $\ell_{y_1}$ is proportionately even higher than its contribution to $\ell_y$. This perhaps should make us realize that this pixel may be less relevant or even irrelevant to label $y$ since it is effectively siding with label $y_1$ (recall Figure 1). We conclude that looking at saliency maps for

---

[1]If function $f$ is computed by a ReLU net with zero bias at each node, then it satisfies $f(\lambda x) = \lambda f(x)$ . Partial differentiation with respect to $\lambda$ at $\lambda = 1$ shows $x \cdot \nabla_x(f) = f(x)$.

non-chosen labels should allow us to fine-tune our estimate of the relevance of a pixel to the chosen label.

Now we formalize the competition idea. Note that positive and negative pixel scores should in general be interpreted differently; the former should be viewed as supporting the chosen label, and the latter as opposing that label.

---

**Competitive version of saliency method**
**Input** : An image, $I$, an underlying saliency method, $S_u$ and some chosen label $y$
**Output** : A saliency map $S$ of same dimensions as the input, $I$

**For** each pixel $p_i$ in the input

1. Calculate the saliency scores assigned by each label to pixel $p_i$ by $S_u$
2. Zero out pixel $p_i$ in the saliency map $S$ if either
   (a) $p_i$ has a positive saliency score for label $y$ that is not maximal among its saliency scores for all labels
   (b) $p_i$ has a negative saliency score for label $y$ that is not minimal among its saliency scores for all labels

---

## 4 WHY COMPETITION WORKS: SOME THEORY

Figure 1 suggests that it is a good idea to zero out some pixels in existing saliency maps, as the existing saliency maps are reflecting information incidental to classification, such as edges. Here we develop a more principled understanding of why adding competition (a) is aggressive enough to zero out enough pixels to help pass sanity checks on randomized nets and (b) not too aggressive so as to retain a reasonable saliency map for properly trained nets.

Adebayo et al. (2018) used linear models to explain why methods like **Gradient⊙Input** fail their randomization tests. These tests turn the gradient into a random vector, and if $\xi_1, \xi_2$ are random vectors, then $x \odot \xi_1$ and $x \odot \xi_2$ are visually quite similar when $x$ is an image. (See Figure 10 in their appendix.) Thus the saliency map retains a strong sense of $x$ after the randomization test, even though the gradient is essentially random. Now it is immediately clear that with $k$-way competition among the labels, the saliency map would be expected to become almost blank in the randomization tests since each pixel is equally likely to receive its highest score from each label, and thus in each saliency map the pixel becomes zero with probability $1 - 1/k$. Thus we would expect that adding competition enables the map to pass the sanity checks in multiclass settings, even for relatively small number of classes. Our experiments later show that the final map is indeed very sparse.

But one cannot use the simple (Adebayo et al., 2018) model to understand why competition yields a reasonable saliency map for properly trained nets. The reason being that the saliency map is not random and depends on the input —since completeness property requires that the sum of pixel saliencies is the label logit. We give a simple model for thinking about this dependence, which we hope will motivate further theory on the best way to aggregate information from different labels. We rely upon the completeness property, although Section 4.1 shows that a weaker property should suffice.

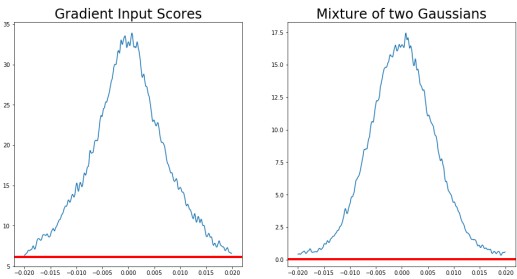

Figure 2: Histogram of Gradient ⊙ Input scores for VGG-19 on Imagenet versus scores produced by mixture of two gaussians.

*Mental model for saliency maps.* One should keep in mind that a saliency map is the outcome of a stochastic process: training a deep net, with stochasticity coming from dataset choice, random initialization, and SGD. Let $\ell_y$ and $\mu_{x,y}$ be random variables denoting respectively the logit value for label $y$ on input $x$ and the corresponding saliency map.

For simplicity we assume the saliency method correlates with a certain ground truth saliency as follows. There exists a subset $S_{x,y}$ of pixels (e.g., the pixels of $x$ that a human may find salient for label $y$) such that for each pixel in $S_{x,y}$ the corresponding coordinate of $\mu_{x,y}$ is distributed as $\mathcal{N}(c, \sigma_1^2)$ and each pixel in $\overline{S_{x,y}}$ is distributed as $N(0, \sigma_2^2)$ where $c = c(x,y)$ and $\sigma_1, \sigma_2$ are constants.

This modeling assumption — "noisy signal" mixed with "white noise"—makes intuitive sense and has some experimental support; see Figure 2 where distribution of saliency scores does seem similar to mixture of two Gaussians (maybe three?), one centered at zero and other at some positive value. One could consider variants of our model, e.g., allowing means and variances to differ across pixels so the maps can include certain image portions (e.g., sharp edges) with very high probability. These don't qualitatively affect the phenomena discussed below.

Now $\mu_{x,y} \cdot \overline{1}$ is the sum of pixel saliencies (where $\overline{1}$ is the all-1's vector), which by completeness, is $\ell_y$. By linearity of expectation $E[\ell_y] = c \cdot |S_{x,y}|$. By measure concentration, with probability at least $1 - \exp(-t^2/4)$ we have

$$|\ell_y - c \cdot |S_{x,y}|| \leq t \left( \sigma_1 \sqrt{|S_{x,y}|} + \sigma_2 \sqrt{|\overline{S_{x,y}}|} \right). \tag{1}$$

Since the sets are fairly large (say, reasonable fraction of all pixels), this concentration should be good[2].

After applying competition, the saliency map changes to $\mu_{x,y} \odot \mathcal{I}_{x,y}$ where $\mathcal{I}_{x,y}$ is the vector with $1$ in coordinates where label $y$ wins the competition for pixels, and $0$ elsewhere. Now we note various plausible conditions under which $\mu_{x,y} \otimes \mathcal{I}_{x,y}$ can be seen as a reasonable approximation to $\mu_{x,y}$. For simplicity we consider how well competition preserves completeness, though obviously other properties can be analysed.

**Theorem 1 (Informal meta-theorem)** *For many plausible conditions on the $S_{x,y}$'s there is a scale factor $\gamma > 0$ (depending on the conditions) such that the vector $\gamma \mu_{x,y} \odot \mathcal{I}_{x,y}$ satisfies completeness up to a small additive error.*

**Proof** (sketch) One uses measure concentration. As an illustrative example suppose $S_{x,y}$'s are disjoint for different $y$'s. For each coordinate in $S_{x,y}$ there is a certain probability $p = p(c, \sigma_1, \sigma_2)$ that the coordinate is nonzero in $\mathcal{I}_{x,y}$. Thus the expected sum of coordinates in the map $\mu_{x,y} \odot \mathcal{I}_{x,y}$ is $p|S_{x,y}|$, and so rescaling coordinates by $c/p$ makes them satisfy completeness, up to an additive error given by measure concentration. Qualitatively similar results are obtained when $S_{x,y}$'s are not disjoint but for $S_{x,y} \cap \cup_{y' \neq y} S_{x,y}$ is small compared to $S_{x,y}$. ∎

*Remarks:*(1) Obviously, the additive error will have smaller impact for the larger logit values than for smaller ones. (2) The above proof suggests that adding competition is akin to sampling a subset of the salient pixels. At least for image data, where pixel-level information has a lot of redundancy, the essence of the original map survives, as seen in the experiments later.

## 4.1 Approximate completeness

While some methods satisfy completeness by design (Bach et al., 2015), others don't. Can our theory apply to the latter class of methods? While an exhaustive study was not performed, randomly sampling a few methods suggests that the saliency maps Ancona et al. (2019) (Shrikumar et al., 2017) in practice satisfy approximate completeness anyway (theory is of course lacking), in sense of the following definition. Figure 3 depicts two saliency methods satisfying approximate completeness.

---

[2]Clearly, weaker but qualitatively similar results hold under changes to the assumptions, e.g. some non-Gaussian distribution for entries of $\mu_{x,y}$; allowing them to be pairwise independent instead of iid, and so on.

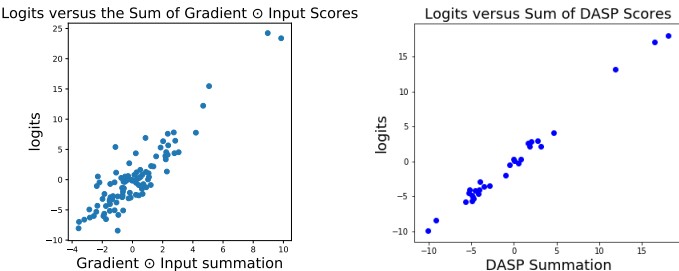

Figure 3: Approximate completeness property of some saliency methods on VGG19, illustrated by near-linear fit of (a) Gradient ⊙ Input on ReLU nets with nonzero bias (b) DASP.

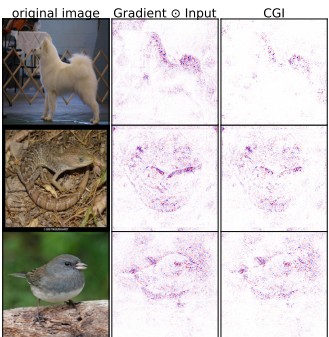

Figure 4: Comparison of CGI saliency maps with Gradient ⊙ Input saliency maps. Original images are shown on the left.

**Definition 2 (Approximate completeness)** *A saliency method satisfies $(\alpha, \beta)$-approximate completeness if sum of the pixel saliencies for label $y$ lies in the interval $[\alpha\ell_y - \beta, \alpha\ell_y + \beta]$.*

**Theorem 3 (Modified meta theorem)** *If a saliency method satisfies $(\alpha, \beta)$-approximate completeness, then under various plausible conditions on the $S_{x,y}$'s similar to Theorem 1 there are constants $(\alpha', \beta')$ such that after applying competition the maps satisfy $(\alpha', \beta')$-approximate completeness.*

## 5 EXPERIMENTS

We consider saliency methods Gradient ⊙ Input, LRP, and DASP, and adding competition to them to get methods CGI, CLRP, and CDASP, respectively. CGI is tested using VGG-19 architecture Simonyan & Zisserman (2015) trained on ImageNet Russakovsky et al. (2015); LRP using VGG-16 architecture with Imagenet; DASP using a CNN model on MNIST. While these architectures are not the most powerful ones around, this is not a limitation for testing saliency methods, which are supposed to work for all architectures.

### 5.1 VISUAL QUALITY OF MAPS

Figure 4 shows underlying saliency maps versus maps with our modifications for VGG-19 on Imagenet. Applying competition does not visibly degrade the map. Some more examples (labeled "original") also appear in Figure 5.

### 5.2 PARAMETER RANDOMIZATION TEST

The goal of these experiments is to determine whether our modification may be applied to an underlying saliency method to pass the first sanity check in Adebayo et al. (2018). We conduct cascading and layerwise randomization as described in Adebayo et al. (2018) from top to bottom.

- The top figure in Figure 5 shows the results of layerwise randomization on Gradient ⊙ Input. (Figure 8 in the Appendix shows the full figure ).The text underneath each image represents which layer of the model was randomized, with the leftmost label of 'original' representing the original saliency map of the fully trained model. The top panel shows the saliency maps produced by **CGI** , and the bottom panel the maps produces by **Gradient ⊙ Input**. We find that the **Gradient ⊙ Input** method displays the bird no matter which layer is randomized, and that our method immediately stops revealing the structure of the bird in the saliency maps as soon as any layer is randomized. Figure 10 in the Appendix shows a similar result but utilizing absolute value visualization. Notice that CGI's sensitivity to model parameters still holds.

- The second figure in Figure 5 shows the results for cascading randomization on Gradient ⊙ Input. The rightmost column represents the original saliency map when all layer weights and biases are set to their fully trained values. The leftmost saliency map represents the map produced when only the softmax layer has been randomized. The image to the right of that when everything up to and including conv5_4 has been randomized, and so on. Again we find that **CGI** is much more sensitive to parameter randomization than **Gradient ⊙ Input**.

- The bottom figure in Figure 5 shows our results for cascading randomization on LRP. We find that our competitive selection process (CLRP) benefits the LRP maps as well. The LRP maps show the structure of the bird even after multiple blocks of randomization, while our maps greatly reduce the prevalence of the bird structure in the images.

- Figure 6 shows our results for cascading randomization on DASP Ancona et al. (2019), a state of the art method that satisfies approximate completeness. DASP still shows the structure of the digit even after randomization, while CDASP eliminates much of this structure.

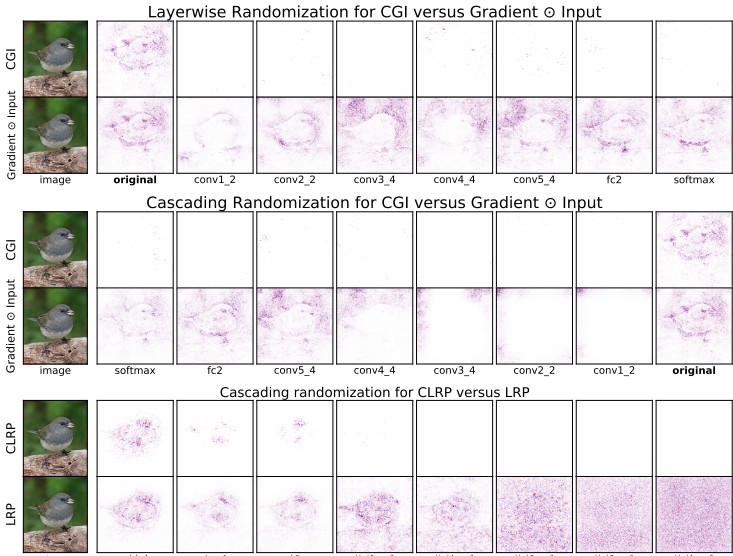

Figure 5: The first two figures depict layerwise and cascading randomization, respectively, for a VGG19 model trained on Imagenet for Gradient ⊙ Input versus CGI. We find that in CGI, the saliency map is almost blank when any weights are reinitialized. By contrast, we find that the original Gradient ⊙ Input method displays the structure of the bird, no matter which layer is randomized. The third figure depicts saliency map for cascading randomization on VGG -16 on Imagenet LRP versus CLRP. We notice that LRP shows the structure of the bird even after multiple blocks of randomization. CLRP eliminates much of the structure of the bird.

## 5.3 DATA RANDOMIZATION TEST

We run experiments to determine whether our saliency method is sensitive to model training. We use a version of Alexnet Krizhevsky et al. (2012) adjusted to accept one color channel instead of three

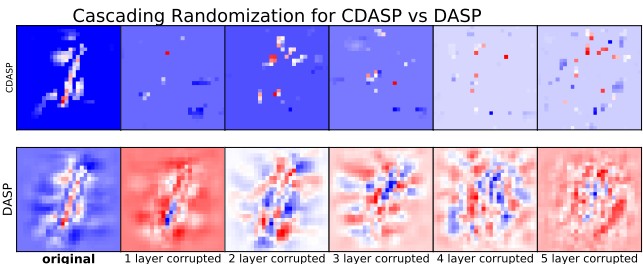

Figure 6: Saliency map cascading randomization on MNIST for CDASP versus DASP. We notice that DASP shows the structure of the digit even after randomization. CDASP eliminates the structure of the digit.

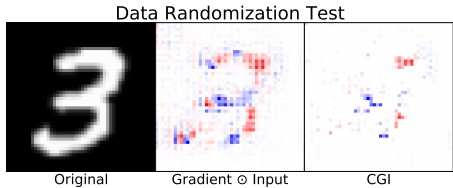

Figure 7: Second sanity check for Alexnet MNIST. On the middle image we find that using the original gradient times input method results in an image where the original structure of the number 3 is still visible. On the right hand side image we find that our modification removes the structure of the original input image, as we would expect for a model that had been fitted on randomized data.

and train on MNIST. We randomly permute the labels in the training data set and train the model to greater than 98 % accuracy and examine the saliency maps. Figure 7 shows our results. On the left hand side is the original image. In the middle is the map produced by **Gradient ⊙ Input** . We find that the input structure, the number 3, still shows through with the **Gradient ⊙ Input** method. On the other hand, **CGI** removes the underlying structure of the number.

## 6 CONCLUSION

*Competition among labels* is a simple modification to existing saliency methods that produces saliency maps by combining information from maps from all labels, instead of just the chosen label. Our modification keeps existing methods relevant for visual evaluation (as shown on three well-known methods **Gradient ⊙ Input**, **LRP**, and **DASP**) while allowing them to pass sanity checks of Adebayo et al. (2018), which had called into question the validity of saliency methods. Possibly our modification even improves the quality of the map, by zero-ing out irrelevant features. We gave some theory in Section 4 to justify the competition idea for methods which satisfy approximate completeness. Many methods satisfy completeness by design, and experimentally we find other methods satisfy approximate completeness.

We hope the simple analysis of Section 4—modeling the saliency map as "noisy signal" mixed with "white noise"—will inspire design of other new saliency maps. We leave open the question of what is the optimum way to design saliency maps by combining information from all labels[3]. When pixel values are spatially correlated it is natural to involve that in designing the competition. This is left for future work.

The sanity checks of Adebayo et al. (2018) randomize the net in a significant way, either by randomizing a layer or training on corrupted data. It is an interesting research problem to devise sanity checks that are less disruptive.

---

[3]One idea that initially looked promising —looking at gradients of outputs of the softmax layer instead of the logits—did not yield good methods in our experiments.

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

## 7 APPENDIX

### 7.1 SOME SALIENCY METHODS

MOVE TO APPENDIX Let $S_y$ denote the logit computed for the chosen output node of interest, $y$.

1. **The Gradient $\odot$ Input explanation**: Gradient $\odot$ Input method Shrikumar et al. (2017) computes $.\frac{\partial S_y}{\partial x} \odot x$ where $\odot$ is the elementwise product.

2. **Integrated Gradients** Integrated gradients Sundararajan et al. (2017) also computes the gradient of the chosen class's logit. However, instead of evaluating this gradient at one fixed data point, integrated gradients consider the path integral of this value as the input varies from a baseline, $\bar{x}$, to the actual input, $x$ along a straight line.

3. **Layerwise Relevance Propagation** Bach et al. (2015) proposed an approach for propagating importance scores called Layerwise Relevance Propagation (LRP). LRP decomposes the output of the neural network into a sum of the relevances of coordinates of the input. Specifically, if a neural network computes a function $f(x)$ they attempt to find relevance scores $R_p^{(1)}$ such that $f(x) \approx \sum_p R_p^{(1)}$

4. **Taylor decomposition** As stated Montavon et al. (2018) for special classes of piecewise linear functions that satisfy $f(tx) = tf(x)$, including ReLU networks with no biases, one can always find a root point near the origin such that $f(x) = \sum_{i=1}^d R_i(x)$ where the relevance scores $R_i(x)$ simplify to $R_i(x) = \frac{\partial f}{\partial x_i} \cdot x_i$

5. **DeepLIFT explanation** The DeepLIFT explanation Shrikumar et al. (2016) calculates the importance of the input by comparing each neuron's activation to some 'reference' activation. Each neuron is assigned an attribution that represents the amount of difference from the baseline that that neuron is responsible for. Reference activations are determined by propagating some reference input, $\bar{x}$, through the neural network.

6. **DASP** The Deep Approximate Shapley Propagation Ancona et al. (2019) is a state of the art method that computes the approximate Shapley values. By Proposition 2 in Ancona et al. (2019) the Shapley values satisfy completeness.

**Relationships between different methods** .Kindermans et al. (2016) and Shrikumar et al. (2017) showed that if modifications for numerical stability are not taken into account, the LRP rules are equivalent within a scaling factor to Gradient $\odot$ Input. Ancona et al. (2018) showed that for ReLU networks (with zero baseline and no biases) the $\epsilon$-LRP and DeepLIFT (Rescale) explanation methods are equivalent to the Gradient $\odot$ Input. DASP Ancona et al. (2019), like DeepLIFT relies on the Shapley values, but designs an efficient way to approximately compute these values.

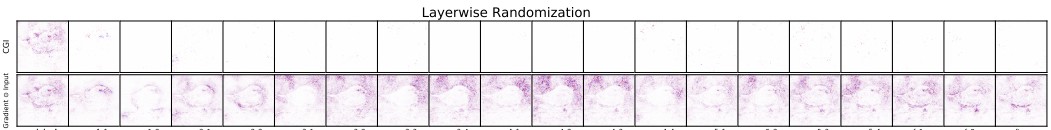

Figure 8: Saliency map for layer-wise randomization of the learned weights. Diverging visualization where we plot the positive importances in red and the negative importances in blue. We find that with CGI, the saliency map is almost blank when any layer is reinitialized. By contrast, we find that Gradient $\odot$ Input displays the structure of the bird, no matter which layer is randomized.

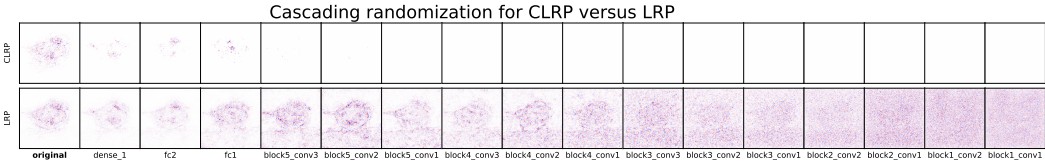

Figure 9: Saliency map cascading randomization LRP versus CLRP.

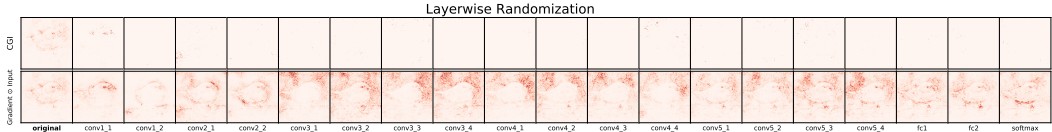

Figure 10: Saliency map for layer-wise randomization of the learned weights. Absolute value visualization where we plot the absolute value of the saliency map. We find that using CGI, the saliency map is almost blank when any layer is reinitialized. By contrast, we find that Gradient $\odot$ Input displays the structure of the bird, no matter which layer is randomized.

