# OpenReview forum: "A Simple Technique to Enable Saliency Methods to Pass the Sanity Checks"
_ICLR.cc/2020/Conference — Reject_

### Official Review · AnonReviewer2 · 2019-10-18
**Official Blind Review #2**

**Rating:** 6

**Review:**

This paper presents a strategy for visualizing activation in networks that corresponds to features in the input layer. It addresses a problem posed for existing methods for characterizing saliency in activation subject to sanity checks which measure the degree to which the activation (saliency) map changes subject to different randomization tests.
The proposed solution involves a simple competition mechanism across saliency maps produced when different logits are considered such that small values are zeroed out in favor of larger values across the logits.
Overall, I find this paper to be interesting and to address a problem worthy of further consideration. While the mechanism for competition is very simple, the resulting activation maps subject to randomization tests are reasonably convincing.
In the ideal case, I would have liked to see different strategies for eliciting competition explored to determine their relative merits. Nevertheless, I expect that such work will follow with this being an initial step in this direction.

**Experience Assessment:**

I have read many papers in this area.

**Review Assessment: Checking Correctness Of Derivations And Theory:**

I assessed the sensibility of the derivations and theory.

**Review Assessment: Checking Correctness Of Experiments:**

I assessed the sensibility of the experiments.

**Review Assessment: Thoroughness In Paper Reading:**

I read the paper at least twice and used my best judgement in assessing the paper.

---

> ### Author Response · Authors · 2019-11-15
> **Response to Official Blind Reviewer 2**
>
> We thank reviewer 2 for their thoughtful review. We agree that the question of how to optimally apply competition among labels remains open.

---

### Official Review · AnonReviewer1 · 2019-10-23
**Official Blind Review #1**

**Rating:** 3

**Review:**

I Summary
The paper directly answers two sanity checks for saliency maps proposed by Adebayo et al (2018): 1. randomizing the weights of a model to prove that the input's resulting saliency map is different from a trained model' saliency map.  2. randomizing the inputs' labels to make the same proof. The authors propose a "competitive version of saliency method" which uses the saliency scores of every pixel for each labels and zero out: positive scored pixels that would not be maximal for the predicted class and negative scored pixels that are not minimal for the worst predicted label.
Overall the method solves the aforementioned sanity checks, the authors claim it also generates more refined saliency maps.

II Comments

1. Content
The paper can be hard to read, due to multiple writing mistakes, abrupt phrasing, not well-articulated sentences. However, the idea is easy to understand and interesting but the contribution does not seem strong enough in its actual state.
My main concern is that the method seems to be designed only to answer the sanity checks: the resulting saliency maps can hardly be seen as more informative as other existing methods (eg figure 1). Quantitative measures (ROAR & KAR, Hooker et al. 2018) or surveys to show that the newly obtained saliency maps are more refined or help to best localize regions of interest would be a big bonus.

2. Writing
The paper comports numerous typos, those do not impact the score of the review except if the sentence is not understandable. Please see the following points as support to improve the clarity of the paper.
- Abstract last sentence: "Some theoretical justification is provided" -> "Some" is vague and makes your claim less credible -> "theoretical justifications are given in the last paragraph to support our method..."
- Intro
   paragraph 2 first sentence lack some words, l 2 product -> a product
   "See paper XX et al" -> "As in XX et al, we can see that" or "As stated in XX et al", "See" is too familiar, formalizing the phrasing gives more credibility to your work
- Related work
  "To give an example, does the map change a lot if we blank out or modify a portion of the image that humans find insignificant Kim et al. (2019)? " This is not very well articulated, "a lot" is vague and a little familiar, "significantly" could be used here. Moreover, the citation is a little abrupt "as we can see in XX" would work better
Little typo on etc..
"fare best" -> far better? The wording is still vague, it would help to add a quantitative measure
that's -> that is
- Section 3
"This figure highlights" -> which figure? (I think you just missed citing the fig here)
- Section 4
First sentence: Why is it a good idea? The claim is a little abrupt and could be detailed a little more
"destroy the saliency map" -> destroy is a very strong word
"These random variables are complicated." -> This statement seems a little out of place and abrupt
"some constants" -> "constants" (too vague otherwise as before)
- Subsection 4.1
"randomly sampling a few such methods" I believe there is a typo?
"See figure 3" is abrupt as a sentence itself "as you can see in figure 3 bla bla"
Figure 4 The image is small and hard to see on printed paper (same for the images in the appendix, they could be stacked over multiple lines instead of just one horizontal row)
Definition 2 punctuation at the end
- Section 5
"The available code for these maps is slow, and computing even gradient for all 1000 ImageNet labels can be rather slow." What is the aim of this sentence?
- subsection 5.3
lables -> labels

III Conclusion
The method itself is interesting, it would be interesting to see more qualitative results on the obtained saliency map itself: Does it produce more information? Is it more meaningful etc. Because as of now, it only seems to answer the two aforementioned sanity checks.  As for the writing, it is not always clear and can impede the understanding of the paper. I would be glad to change my review if those points are addressed.


**Experience Assessment:**

I have read many papers in this area.

**Review Assessment: Checking Correctness Of Derivations And Theory:**

I did not assess the derivations or theory.

**Review Assessment: Checking Correctness Of Experiments:**

I assessed the sensibility of the experiments.

**Review Assessment: Thoroughness In Paper Reading:**

I read the paper at least twice and used my best judgement in assessing the paper.

---

> ### Author Response · Authors · 2019-11-15
> **Response to Official Blind Reviewer 1**
>
> We thank reviewer 1 for their thoughtful review.
>
> We apologize for the typos, and are correcting them in the revised version. As for the sentence beginning Section 4, we attempted to show, in Figure 1, that that although nearly all the pixels in the actual digit ‘3’ are highlighted in the saliency map with chosen label 3, some of these pixels appear more relevant for other classes, for example in the map for logit 7 the top and backbone of the 3 can clearly be seen.  This suggests that just because a pixel is present in the saliency map for logit ‘3’, does not mean that is it primarily indicative of the label being ‘3’, especially if it assigns a lower score to the label being ‘3’ than to ‘7’.

---

### Official Review · AnonReviewer4 · 2019-11-03
**Official Blind Review #4**

**Rating:** 3

**Review:**

Summary:

The paper proposes a simple technique to address the problem introduced by Adebayo et al. that several saliency approaches do not pass sanity checks. The proposed approach computes the saliency maps for all the classes and removes the pixels that play a role in predicting several classes.

Strengths:

1. Simple and intuitive approach.
2. Well written and easy to read paper.
3. The introduced approach makes Grad.Input pass the sanity checks introduced by Adebayo et al.

Weaknesses:

1. For any interpretability technique, passing the sanity check is a must, but just because a saliency technique passes the sanity checks, it doesn’t mean that these maps explain the network’s decision well.
2. Lack of any quantitative evaluation (such as localization or pointing experiment) of their approach.
3. Failure to show if the resultant maps are class-discriminative. Show performance on images with multiple classes.
4. In fig 1,  In Grad . Input, I see positive values or negative values even when the original pixels are not active. This doesn’t explain the presence of edges causing high values in the G.I map for such pixels, right?
5. In figure 1, These maps only assign values to the pixels that need to be removed to make a certain classification decision. The regions that need to be active but are not present are not highlighted.
6. In figure 1 the shown CGI Map is for which class?
7. So, is the approach only applicable to such systems where the completeness is true? Can the authors provide a list of approach that satisfy completeness:
8. Page 3 last paragraph: Consider the example in figure 1. Let's consider the maps for digit 3 and 5. For the top horizontal part of the digit, it plays a role in determining both 3 and 5. Assume that for one such pixel the value of h_5_i is greater thatn h_3_i (looking at the figure it is not unreasonable to expect that). Just because the g.input value of h_5_i is greater that h_3_i , are the authors saying that the top part is irrelevant?
9. How does CGI look for the original 3 on standard model?
10. Could the authors provide more intuition as the why the gradients of outputs from softmax layer doesn’t give good results? The proposed approach from https://arxiv.org/pdf/1908.04351.pdf suggests that computing gradients from last layer improves the class discriminative behaviour.



**Experience Assessment:**

I have published one or two papers in this area.

**Review Assessment: Checking Correctness Of Derivations And Theory:**

I assessed the sensibility of the derivations and theory.

**Review Assessment: Checking Correctness Of Experiments:**

I carefully checked the experiments.

**Review Assessment: Thoroughness In Paper Reading:**

I read the paper at least twice and used my best judgement in assessing the paper.

---

> ### Author Response · Authors · 2019-11-15
> **Response to Official Blind Reviewer 4**
>
> We thank reviewer 4 for a  thoughtful review.
>
> 4. In Figure 1, the data was normalized before feeding it to the neural network, so the background values are not all zero. We would also like to note that it is possible that any anomalous values in the input image may propagate spuriously to the logit, whether they be edges or some other image feature.
> 6. Figure 1 shows the CGI map for chosen logit ‘3’.
> 7. In order to compare across logits, either completeness or approximate completeness (meaning there is a correlation between the sum of saliency scores for each logit and the logit value) must be satisfied. LRP, gradient * input for ReLU nets with no bias, DeepLift for ReLU nets with no bias, all satisfy completeness. Gradient *input seems to satisfy approximate completeness even for ReLU networks with bias.
> 8. The authors do not mean to imply that such ‘shared features’ (those indicative for multiple classes) are irrelevant. However, our competition idea forces each pixel to  choose one label (the one giving it the maximal score). While at first glance this seems to discard potentially relevant pixels, the theory suggests reasons why it doesn’t happen too often.
> 9. This may be seen in Figure 1 in the rightmost column.
> 10. Taking the gradient of the post-softmax probability with respect to the input does not satisfy completeness for the logit value for gradient * input, and we found it did not pass the sanity checks well. The referenced paper computes a different calculation, which is using the derivative of the post softmax probability with respect to the logit as the initial value for LRP. We did not test this method, as it is more specific to LRP and we were examining a more broad class of saliency methods.

---

### Decision · Program_Chairs · 2019-12-19

**Decision:**

Reject

**Comment:**

This submission proposes a method to pass sanity checks on saliency methods for model explainability that were proposed in a prior work.

Pros:
-The method is simple, intuitive and does indeed pass the proposed checks.

Cons:
-The proposed method aims to pass the sanity checks, but is not well-evaluated on whether it provides good explanations. Passing these checks can be considered as necessary but not sufficient.
-All reviewers agreed that the evaluation could be improved and most reviewers found the evaluation insufficient.

Given the shortcomings, AC agrees with the majority recommendation to reject.